# Efficacy and Safety of the *Cudrania tricuspidata* Extract on Functional Dyspepsia: A Randomized Double-Blind Placebo-Controlled Multicenter Study

**DOI:** 10.3390/jcm10225323

**Published:** 2021-11-16

**Authors:** Jinyoung Shin, Tae-Hoon Oh, Joo-Yun Kim, Jae-Jung Shim, Jung-Lyoul Lee

**Affiliations:** 1Department of Family Medicine, Research Institute on Healthy Aging, Konkuk University School of Medicine, Seoul 05030, Korea; 2Division of Gastroenterology, Vievis Namuh Hospital, Seoul 06117, Korea; taehoonoh@vievisnamuh.com; 3R&BD Center, hy. Co., Ltd., Yongin 17086, Korea; monera@hy.co.kr (J.-Y.K.); jjshim@hy.co.kr (J.-J.S.); jlleesk@hy.co.kr (J.-L.L.)

**Keywords:** *Cudrania tricuspidata*, dyspepsia, clinical trials

## Abstract

*Cudrania tricuspidata* is a folk remedy used to treat inflammation in patients with tumors or liver damage. This study investigated the efficacy of *Cudrania tricuspidata* extract (CTE) for relieving the symptoms of functional dyspepsia. In an 8-week, randomized, double-blind, placebo-controlled study, 100 adults with any condition featured in the Rome IV criteria and a Gastrointestinal Symptoms Scale (GIS) score ≥4 were randomly allocated to take either a placebo (maltodextrin) or a 50 mg CTE tablet, which equally included celluloses, magnesium stearate, and silicon dioxide, twice daily, 20 January 2020, and 3 August 2020. Among the 83 participants finally analyzed, the CTE group was associated with a significant reduction in the gastrointestinal symptom rating scale (day 0: 8.0 ± 5.2, day 28: 4.7 ± 3.9, and day 56: 2.3 ± 2.4, *p* < 0.001, respectively) in comparison with the control group (day 0: 8.1 ± 4.7, day 28: 7.8 ± 5.7, and day 56: 7.5 ± 6.6, *p* > 0.05) after adjusting for smoking, drinking, eating habits, stress levels, and caffeine intake. The CTE group resulted in significant improvements of GIS, Nepean Dyspepsia Index (Korean version), and functional dyspepsia-related quality of life over time. There were no different adverse events (*p* = 0.523). These findings suggest that CTE is safe and efficacious for alleviating gastrointestinal symptoms in patients with functional dyspepsia.

## 1. Introduction

Functional dyspepsia (FD) is a complex of symptoms that occur in the gastroduodenal region of the gastrointestinal tract and include epigastric pain, burning, postprandial fullness, or early satiety without structural problems [1]. The prevalence can vary based on the definition or diagnostic criteria used, but approximately 16% of healthy individuals in the general population are affected [1]. The symptom-based criteria in current use for FD are the Rome IV criteria (fourth edition), developed by a group of experts in functional gastrointestinal disorders [2]. FD is a chronic health concern; therefore, it affects the quality of life and social functioning [3]. FD management includes eradicating Helicobacter pylori if the infection is present, acid suppression therapy, prokinetic drugs, and central neuromodulators. Some evidence suggests that lifestyle changes or exercise can lead to symptom improvement. A systematic review of 16 studies examining the effect of nutrients, food, and food components found that a diet with reduced wheat and dietary fats may improve FD symptoms [4]. As few randomized controlled trials of dietary manipulation exist, empirical pharmacological therapy is the current treatment strategy [1,5]. As the diagnosis of FD must exclude organic causes of various symptoms, little evidence exists to corroborate the effectiveness of treatments, and none are proven to alter the long-term natural history of FD [1,6]. Nevertheless, the increased risk of developing a neuroendocrine tumor in patients who have used a proton pump inhibitor for more than ten years or at higher doses was reported [7]. Therefore, there has been an attempt to apply complementary therapy in the treatment of FD [8].

The cortex, leaf, and root bark of *Cudrania tricuspidata* have been used in folk remedies in South Korea, China, and Japan to treat inflammation [9]. Several studies have reported the effects of *Cudrania tricuspidata* extract (CTE) in anti-tumorigenesis, the inhibition of oxidative stress-induced liver injury, and protection from reflux esophagitis [10,11,12]. A previous study found that CTE reduced gastric acid secretion by reducing H2-receptor activity and increasing mucin genes to protect the gastric mucosa in a rat model [13]. Changes in mucin gene expression may affect FD [14]. Therefore, we hypothesized that the pharmacological actions of CTE might have favorable therapeutic efficacy in FD patients.

This study was designed to determine whether an extract of *C. tricuspidata* leaves would improve gastrointestinal symptoms, such as epigastric pain, acid reflux, heartburn, nausea, and vomiting, compared with a placebo.

## 2. Materials and Methods

### 2.1. Study Design

This was an 8-week randomized double-blind placebo-controlled trial evaluating the efficacy and tolerability of CTE among FD patients. This study was conducted on outpatients from two hospitals, a university hospital and a hospital specializing in the gastrointestinal tract, in Seoul, Korea. The study protocol was approved by the Institutional Review Board (IRB file no. KUMC 2019-11-010) and registered with the Clinical Research Information Service (CRIS, no. KCT0005020) in the Korea Disease Control and Prevention Agency supported by the Ministry of Health and Welfare before the time of first patient enrollment. CRIS joined the WHO International Clinical Trials Registry Platform (ICTRP) as the 11th member of Primary Registry. All the methods were carried out following the relevant guidelines and regulations.

We undertook a priori power analysis to estimate the required sample size as a superiority test. A previous study found that gastrointestinal symptoms in the treatment group were associated with a decreased score compared to the initial scores (*n* = 30, mean −4.7, standard deviation 4.6) and placebo group (*n* = 30, mean −1.7, standard deviation 4.6) [15]. Considering 1:1 allocation, a power of 80%, and a type 1 error rate (alpha) of 5%, at least 37 participants in the experimental group were calculated as the appropriate number of participants in the study. The total number of participants to establish an effect was estimated to be 100, considering a 25% drop-out rate.

Volunteers were screened after completing a signed informed consent. On visit 1, demographic data (age, birth date, and sex), comorbidity (diagnosis by medical doctors), surgical history within six months, and medication type were surveyed. Data on lifestyle, such as physical activity (none/1–2/3–4/≥5 days per week), smoking (non-smoker, ex-smoker/current smoker), alcohol consumption (more or less than once a month), stress level (none/mild/moderate/severe), caffeine intake (mean cups per week in a drink and mean gram per week in chocolate), and dietary habits (regular meal: yes/no, mealtime: <10/10–20/≥20 min, and overeating times per week) were obtained. In a fasting state, blood tests (erythrocyte sedimentation rate, ESR; C-reactive protein, CRP; aspartate aminotransferase, AST; alanine aminotransferase, ALT; thyroid-stimulating hormone, TSH; creatinine, creatinine kinase, CK; and gamma-glutamyl transpeptidase, *r*-GTP), urine tests, electrocardiography, and pregnancy tests in females were undertaken. We examined height, weight, blood pressure, and pulse rate, and body mass index (BMI) was calculated as weight divided by height squared. To ensure that abnormalities in the digestive system did not cause discomfort, an endoscopy was conducted before the study if one was not implemented within the last ten months.

On visit 2 (baseline/day 0), information on diet habits by 24 h recall tests and questionnaires for study outcomes was obtained, and eligible participants were randomized. Enrolled participants were assigned to either the CTE group or control group by block randomization using the SAS^®^ system. We kept the master randomization list with the details of allocation safely and confidentially with the sponsor. At visits 3 (day 28) and 4 (day 56), we took vitals and conducted a physical examination, counted returned pills, provided new pills, rated the gastrointestinal symptoms questionnaires, and recorded any adverse events. At visit 4 (last visit, day 56), we also collected a blood sample to assess ESR and CRP.

### 2.2. Participants

Those with gastrointestinal symptoms and who wished to participate were informed about the study. If agreeable, they were assessed by the principal investigator for eligibility based on the following inclusion/exclusion criteria. We registered participants when all conditions were met: adults aged between 20 and 70 years; one gastrointestinal symptom, such as bloated after meals, early satiety, epigastric area pain, burning in the upper abdomen, which started at least six months before the study and has been occurring for three months according to the Rome IV criteria; without evidence of structural lesions in the upper gastrointestinal system on endoscopy; at least four symptoms with moderate or severe intensity among ten items of the gastrointestinal symptom scale (GIS) in the last two weeks; agreement to participate in the study and a signed written consent form.

Patients were excluded from consideration if they had malignancy, stroke, or cardiovascular disease that was being treated or not being controlled. Participants were ineligible for participation in this study if they were pregnant, lactating, or had a history of ulcers in the last six months, gastrointestinal surgery, heavy drinkers (male 14 units and female 7 units), or taking an H2-receptor inhibitor, corticosteroids, non-steroidal anti-inflammatory drug, or aspirin. Individuals with uncontrolled hypertension (systolic blood pressure ≥160 mmHg or diastolic blood pressure ≥100 mmHg), uncontrolled diabetes mellitus (fasting glucose level ≥180 mg/dL), abnormal results, creatinine (≥2 times the standard upper limit), AST or ALT (≥3 times the standard upper limit), amylase or lipase (≥2 times the standard upper limit), TSH (≤0.1 μIU/mL or ≥10 μIU/mL), or known hypersensitivity to experimental agents were also excluded from participating in the study.

### 2.3. Interventions

We used tablets containing either 50 mg of CTE or placebo (maltodextrin) for the intervention and control groups for 8 weeks, respectively. Other ingredients were included equally (microcrystalline cellulose 279 mg, calcium carboxymethyl cellulose 5.25 mg, magnesium stearate 7.0 mg, silicon dioxide 5.25 mg, hydroxypropyl methylcellulose 3.50 mg). These were identical in appearance, shape, color, and packaging. Participants were instructed to take one tablet twice daily. Based on the previous study, CTE inhibited gastric mucosal damage in a dose-dependent manner, with the most significant reduction at 10 mg/kg [13]. We calculated the human equivalent dose from animal doses [16].

Substances that could affect the health of the stomach were prohibited from the screening visit to the closing visit. Participants informed the researcher immediately if they took any medications.

### 2.4. Outcome Measures

#### 2.4.1. Primary Outcome Measure: Gastrointestinal Symptom Rating Scale

The Gastrointestinal Symptom Rating Scale (GSRS) is a disease-specific instrument that includes 15 items addressing different gastrointestinal symptoms; upper abdominal symptoms, such as abdominal pain, heartburn, acid regurgitation, sucking sensation, nausea or vomiting, borborygmus, abdominal distension, and eructation; and lower abdominal symptoms, such as increased flatus, decreased or increased passage of stools, loose stools, hard stools, urgent need for defecation, and feeling of incomplete evacuation [17]. A 4-point Likert type scale, from zero (absence of bothersome symptoms) to three (very bothersome symptoms), was rated. The Korean version of GSRS was validated and modified [18]. We assessed the upper abdominal symptom scores and the total scores of GSRS at days 0, 28, and 56.

#### 2.4.2. Secondary Outcomes: The Gastrointestinal Syndrome Scale, Nepean Dyspepsia Index, and FD-Related Quality of Life

The GIS includes 10 items typically related to FD: epigastric or upper abdominal pain, abdominal cramps, bloating, early satiety, loss of appetite, sickness, nausea, vomiting, retrosternal discomfort, and acidic eructation/heartburn [19]. The intensity of each item was rated on a 5-point Likert scale (0 = no problem, 1 = mild problem, 2 = moderate problem, 3 = severe problem, and 4 = very severe problem), and higher scores represented severe disease. The GIS can be evaluated quickly (2–3 min), which is helpful for clinical trial validation and monitors treatment effects. Initially, subjects with four out of 10 symptoms and a total score of 12 or more could participate in this study; then, we assessed GIS at days 28 and 56.

The Korean version of the Nepean Dyspepsia Index (NDI-K) is a validated tool for evaluating clinically meaningful FD changes, gastrointestinal symptoms, and the effects on health-related quality of life [20]. We measured symptom-based questions regarding the frequency, severity, and degree of distress among 15 symptoms over the prior two weeks, including pain in the upper abdomen, discomfort, burning, heartburn, cramps, chest discomfort, inability to finish a regular meal, bitter-tasting fluid in the mouth, fullness after eating, pressure in the upper abdomen, bloating, nausea, belching, vomiting, and bad breath [21]. The frequency is measured using a 5-point Likert scale (0 = none, 1 = 1–4 days, 2 = 5–8 days, 3 = 9–12 days, and 4 = daily or almost daily). The severity is measured using a 6-point Likert scale (0 = not at all or not applicable, 1 = very mild, 2 = mild, 3 = moderate, 4 = considerable, and 5 = extreme). The degree of distress is measured using a 5-point Likert scale (0 = not at all, 1 = mild, 2 = moderate, 3 = considerable, and 4 = extreme). The total scores were zero to 195, the higher scores indicating a worse status in patients with FD. We assessed NDI-K at days 0, 28, and 56.

FD-related quality of life (FD-QoL) is a reliable and valid measurement for health-related quality of life. It is used to evaluate the effectiveness of treatments in patients with FD [22]. FD-QoL consists of four dimensions and 21 items, the dimensions being psychological (six items), role-functioning (six items), eating (five items), and liveliness (four items). A 5-point Likert scale was used to rate each item (0 = not at all, 1 = mild, 2 = moderate, 3 = considerable, and 4 = extreme) [22]. We assessed FD-QoL at days 0 and 56.

### 2.5. Side Effects

Side effects were recorded according to the medical dictionary for regulatory activities at every visit [23]. We assessed the severity and the relation with CTE.

### 2.6. Statistical Analysis

The effectiveness was analyzed only for subjects who finished this clinical trial and had no significant violations. Demographic variables for continuous variables were compared using a *t*-test or Wilcoxon rank-sum test between the control and CTE groups. The categorical variables were compared using a chi-square test or Fisher’s exact test. We selected the appropriate analysis method according to the normality of the group data by the Shapiro–Wilk normality test. In comparing outcome variables within each group for the changes from baseline, a paired *t*-test was used at days 28 and 56. A two-sample *t*-test was used in the comparison between groups at days 0, 28, and 56, and the generalized linear model was used in the comparison between groups from day 0 to 56 after adjustments for age, sex, BMI, smoking, physical activity, alcohol consumption, stress, diet habit, and caffeine intake. The incidence of side effects was calculated and compared using the chi-square test or Fisher’s exact test. The blood and urine test results, weight, pulse rate, and blood pressure were paired with a *t*-test within groups, and a two-sample *t*-test or Wilcoxon rank-sum test between groups. Electrocardiography was compared between baseline and day 56 in both groups according to normal or abnormal results by a McNemar test. The safety analysis was performed on all enrolled subjects after randomization (total *n* = 100). For all the tests, statistical significance was set at *p* < 0.05 (two-tailed). All data were analyzed using SAS^®^ (v9.4, SAS Institute, Cary, NC, USA).

## 3. Results

### 3.1. Demographics

A total of 100 participants were finally enrolled after excluding 15 people in the screening, and 83 participants were included in this analysis. During the trial, 17 participants (11 in the CTE group and six in the control group) were excluded from the final analysis (Figure 1).

Of these, two participants in the CTE group withdrew their consent; others were excluded due to intake of a prohibited medication in the CTE (*n* = 1) and control (*n* = 3) groups, or low compliance with drug intake (<80%) in the CTE (*n* = 8) and control (*n* = 3) groups. The compliance of this trial in the CTE group (99.5 ± 9.9%) was similar to the control group (96.5 ± 8.4%) (*p* = 0.133).

Demographic characteristics are shown in Table 1, which were homogenous with no statistically significant differences between the control group and the CTE group.

### 3.2. Primary and Secondary Outcomes

Changes in the total GSRS scores, GIS, NDI-K, FD-QoL, ESR, and CRP across the two groups and visits are detailed in Table 2. The frequency scales of the NDI-K were represented in Appendix A. The changes in each symptom score of GSRS are represented in Figure 2.

All symptoms at day 0 were no different between the two groups. The *p*-value was calculated using a paired *t*-test to compare within groups from baseline and day 56 or a generalized linear model to compare between groups after adjusting for age, sex, body mass index, smoking, physical activity, alcohol consumption, stress, diet habit, and caffeine intake.

As a primary outcome, a significant reduction in total GSRS scores in the CTE group was observed on days 28 (8.0 ± 5.2 to 4.7 ± 3.9, *p* < 0.001) and 56 (2.3 ± 2.4, *p* < 0.001) compared to the control group (8.1 ± 4.7 to 7.8 ± 5.7 in day 28, *p* = 0.679; 7.5 ± 6.6 in day 56, *p* = 0.339). All symptoms among the GSRS in the CTE group were improved compared to the control group with the significant differences of between-group in total GSRS score (*p* < 0.001) and all symptoms (*p* < 0.01). In the CTE group, GIS and NDI-K scores were reduced at day 28 and maintained at day 56. The final score of FD-QoL was significantly decreased in the CTE group compared to the initial score. Differences between groups were found after adjusting for confounding factors (*p* < 0.001).

Pre- and post-blood tests (CRP, AST, ALT, TSH, CK, and r-GTP) confirmed no statistical significance in both groups. However, the level of ESR in the CTE group was decreased compared to the control group (*p* = 0.006). Blood pressure, pulse rate, weight, and electrocardiography had no statistical significance in either group (*p* > 0.05).

### 3.3. Safety Analysis

Side effects in this trial are represented in Table 3. Side effects were reported by 27 of the 100 subjects (27%, 12 cases in the CTE group, and 15 cases in the control group). The most common was nausea (4%). No significant differences in side effects were found between the CTE and control groups.

All side effects were mild to moderate in severity and resolved spontaneously without further treatment or sequelae. No subject discontinued participating in this study due to side effects. Among the reported side effects, the rate possibly related to this intervention was 8.33% (1/12) in the CTE group and 26.67% (4/15) in the control group (*p* = 0.523), which were only gastrointestinal symptoms, and not hypersensitivity or allergic reactions.

## 4. Discussion

This study confirmed that the intake of CTE in relatively healthy Korean adults was associated with a statistically significant reduction in gastrointestinal symptoms compared to the placebo, after adjusting for confounding factors including smoking, drinking, eating habits, stress levels, and caffeine intake, which were expected to affect gastrointestinal symptoms. In the primary outcomes of this study, the total score and all scores of upper abdominal symptoms in GSRS were reduced in the intervention group. Furthermore, this change was found at day 28 and maintained until the end of day 56. We also found the CTE was associated with an improvement in health-related quality of life in subjects with FD. The improvement of upper abdominal symptoms (stomach pain, heartburn, acid reflux, deep licking, nausea, vomiting, abdominal pain, trimming, satiety, bloating, etc.) as well as lower abdominal symptoms (increased flatus, stool passage, or consistency) was an opportunity to confirm the extended effect of CTE on the symptoms of the esophagus and stomach-related upper abdomen through a previous animal study [11,12,13]. CTE was well tolerated with no significant side effects or blood results over time.

The favorable effect of CTE may inhibit H2-receptor-mediated cyclic adenosine monophosphate (AMP) production and gastric acid secretion in FD patients [12]. Although there was no difference in CRP, a reduction in ESR, as a marker for acute phase inflammation, was confirmed in this study. This could suggest a possible mechanism that affects low-grade mucosal inflammation in FD [1,24]. However, further study is needed to identify the mechanisms associated with the improvement.

In our study, a difference in gastrointestinal symptom effects between CTE and placebo not only appeared after four weeks of administration, but the gap was also increased after 8 weeks of further administration. A change of at least 10 points on the NDI total scale corresponded to a clinically meaningful improvement in the patient [25]. We confirmed the efficient improvement of symptoms in the CTE group (total mean change: −33.7 ± 28.6 in 4 weeks and −50.1 ± 36.3 in 8 weeks) compared to a limited change in the control group. The effectiveness comparable to the decrease in the NDI-K score with Mosapride, a selective 5-HT_4_ agonist, was shown [21]. The improvement in symptoms using the GIS survey used in selecting the participants was also identified in the control group. However, we can find a significant difference when comparing the groups.

The limitations of this study are the relatively short duration, small sample size, and lack of dose–response data. The enrollment and evaluation of subjects using a self-reported questionnaire might have resulted in biased study findings [26]. Since the diagnostic criteria for FD were various and the treatment of FD is not clear as a single agent, the effect of CTE could not be compared with the existing treatment but was compared with placebo. Despite these limitations, this is the first research to demonstrate the efficacy and safety of CTE in the treatment of FD patients.

## 5. Conclusions

These findings provide clinical evidence to support the efficacy of CTE in alleviating gastrointestinal symptoms in patients with FD. No serious side effects were observed, thus keeping the pharmacological safety of CTE, which can be considered as a viable candidate for the management of FD. Further studies addressing the effect of CTE should be considered in the long term and at a large scale, dose-dependent or compared with current treatment methods.

## Figures and Tables

**Figure 1 jcm-10-05323-f001:**
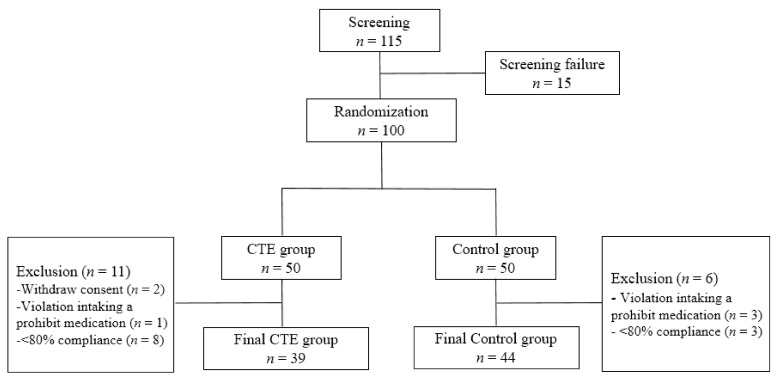
Flow chart of the study population.

**Figure 2 jcm-10-05323-f002:**
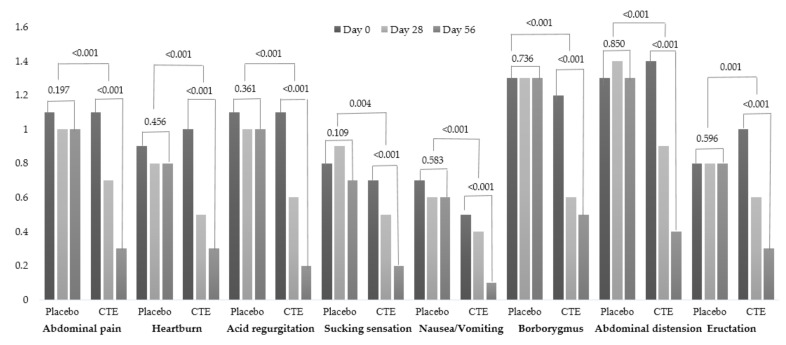
The changes of upper abdominal symptom scores in GSRS.

**Table 1 jcm-10-05323-t001:** Baseline characteristics of study participants.

Characteristics	Intervention Group (*n* = 39)	Control Group (*n* = 44)	Total (*n* = 83)	*p*-Value
Age, years	39.9 ± 12.7	38.0 ± 12.5	38.9 ± 12.5	0.368 ^a^
Sex				0.113 ^b^
Male	20 (51.3)	15 (34.1)	35 (42.2)	
Female	19 (48.7)	29 (65.9)	48 (57.8)	
Body mass index (kg/m^2^)	23.2 ± 3.5	24.4 ± 4.1	23.8 ± 3.8	0.266 ^a^
Physical activity				0.160 ^c^
≥5 days/week	1 (2.6)	4 (9.1)	5 (6.0)	
3–4 days/week	17 (43.6)	9 (20.5)	26 (31.3)	
1–2 days/week	10 (25.6)	13 (29.5)	23 (27.7)	
None	11 (28.2)	18 (40.9)	29 (34.9)	
Smoking				0.472 ^c^
Non-smoker	29 (74.4)	33 (75)	62 (74.7)	
Ex-smoker	3 (7.7)	1 (2.3)	4 (4.8)	
Current smoker	7 (17.9)	10 (22.7)	17 (20.5)	
Alcohol drinking				0.773 ^c^
≥one-time/month	32 (82.1)	35 (79.5)	67 (80.7)	
<one-time/month	7 (17.9)	9 (20.5)	16 (19.3)	
Perceived stress				0.849 ^c^
None	1 (2.6)	1 (2.3)	2 (2.4)	
Mild	20 (51.3)	19 (43.2)	39 (47.0)	
Moderate	16 (41.0)	20 (45.5)	36 (43.4)	
Severe	2 (5.1)	4 (9.1)	6 (7.2)	
Caffeine intake				
Drink, mean cup/week	7.9 ± 10.9	7.3 ± 12.7	7.6 ± 11.8	0.757 ^d^
Chocolate, mean gram/week	70 ± 105	42 ± 70	56 ± 91	0.054 ^d^
Regular meal				0.805 ^b^
Yes	22 (56.4)	26 (59.1)	48 (57.8)	
No	17 (43.6)	18 (40.9)	35 (42.2)	
Meal time				0.558 ^c^
<10 min	7 (17.9)	11 (25.0)	18 (21.7)	
10–20 min	24 (61.5)	28 (63.6)	52 (62.7)	
≥20 min	8 (20.5)	5 (11.4)	13 (15.7)	
Overeating				0.893 ^b^
<3 times/week	28 (71.8)	31 (70.5)	59 (71.9)	
≥3 times/week	11 (28.2)	13 (29.5)	24 (28.9)	
Comorbidity				0.429 ^b^
Yes	17 (43.6)	23 (52.3)	40 (48.2)	
No	22 (56.4)	21 (47.7)	43 (51.8)	
Medication				0.344 ^b^
Yes	19 (48.7)	26 (59.1)	45 (54.2)	
No	20 (51.3)	18 (40.9)	38 (45.8)	

Values are presented as mean ± standard deviation or number (%). ^a^: *p*-value for two-sample *t*-test, ^b^: *p*-value for Chi-square test, ^c^: *p*-value for Fisher’s exact test, ^d^: *p*-value for Wilcoxon rank-sum test.

**Table 2 jcm-10-05323-t002:** Primary and secondary outcomes of this study to evaluate the efficacy of CTE.

		*n*	Day 0	Day 28		Day 56		*p*-Value **
			Mean (SD)	Mean (SD)	*p*-Value *	Mean (SD)	*p*-Value *	
GSRS	Control	44	8.1 (4.7)	7.8 (5.7)	0.679	7.5 (6.6)	0.339	<0.001
	Intervention	39	8.0 (5.2)	4.7 (3.9)	<0.001	2.3 (2.4)	<0.001	
	*p*-value ^†^		0.969	<0.001		<0.001		
GIS	Control	44	19.3 (6.2)	16.0 (7.9)	<0.001	14.6 (9.8)	<0.001	<0.001
	Intervention	39	19.7 (7.4)	10.8 (5.4)	<0.001	5.9 (4.3)	<0.001	
	*p*-value ^†^		0.934	<0.001		<0.001		
NDI-K	Control	44	64.5 (28.8)	58.6 (34.6)	0.108	58.9 (43.2)	0.268	<0.001
	Intervention	39	69.1 (35.5)	35.4 (22.2)	<0.001	19.0 (13.5)	<0.001	
	*p*-value ^†^		0.781	<0.001		<0.001		
FD-QoL ^‡^	Control	44	30.4 (13.0)	-	-	27.2 (19.0)	0.166	<0.001
	Intervention	39	32.2 (15.5)	-	-	13.0 (10.9)	<0.001	
	*p*-value ^†^		0.568	-		<0.001		
ESR(mm/hr)	Control	44	9.64 (9.86)	-	-	11.32 (11.44)	0.196	0.006
	Intervention	39	6.77 (6.05)	-	-	5.79 (4.17)	0.076	
	*p*-value ^†^		0.264	-		0.017		
CRP(mg/dL)	Control	44	0.14 (0.25)	-	-	0.19 (0.51)	0.601	0.770
	Intervention	39	0.09 (0.05)	-	-	0.09 (0.06)	0.881	
	*p*-value ^†^		0.716	-		0.972		

CTE: *Cudrania tricuspidata* extract, SD: standard deviation, GSRS: Gastrointestinal Symptom Rating Scale, GIS: Gastrointestinal Symptoms Scale, NDI-K: Nepean Dyspepsia Index—Korean Version, FD-QoL: functional dyspepsia-related quality of life, ESR: erythrocyte sedimentation rate, CRP: C-reactive protein. * Compared within groups for changes from baseline; *p*-value for paired *t*-test, ** Compared between groups in this study; *p*-value for the generalized linear model adjusted by age, sex, BMI, smoking, physical activity, alcohol consumption, stress, diet habit, and caffeine intake, ^†^ Compared between groups: *p*-value by two-sample *t*-test. ^‡^ Lower score means better status.

**Table 3 jcm-10-05323-t003:** Side effects in this study.

	Intervention Group (*n* = 50)		Control Group (*n* = 50)		Total (*n* = 100)		*p*-Value
	Case	%	Case	%	Case	%	
Total side effects	12	24.0	15	30.0	27	27.0	0.461
Epigastric pain	1	2.0	2	4.0	3	3.0	
Abdominal distension	0	0.0	3	6.0	3	3.0	
Constipation	1	2.0	0	0.0	1	1.0	
Diarrhea	1	2.0	0	0.0	1	1.0	
Dyspepsia	0	0.0	1	2.0	1	1.0	
Nausea	2	4.0	2	4.0	4	4.0	
Chest pain	2	4.0	0	0.0	2	2.0	
Hordeolum	1	2.0	1	2.0	2	2.0	
Upper respiratory tract infection	0	0.0	1	2.0	1	1.0	
Myalgia	2	4.0	1	2.0	3	3.0	
Headache	0	0.0	2	4.0	2	2.0	
Influenza A virus test positive	0	0.0	1	2.0	1	1.0	
Allergic conjunctivitis	1	2.0	0	0.0	1	1.0	
Allergy to chemicals	1	2.0	0	0.0	1	1.0	
Hypersensitivity	0	0.0	1	2.0	1	1.0	
Mild	11	91.67	12	80.00	23	85.19	0.605
Moderate	1	8.33	3	20.00	4	14.81	
Severe	0	0.00	0	0.00	0	0.00	
Possibly related *	1	8.33	4	26.67	5	18.52	0.523
Probably not related	4	33.33	5	33.33	9	33.33	
Definitely not related	7	58.33	6	40.00	13	48.15	
Unknown	0	0.00	0	0.00	0	0.00	

* All events were related with gastrointestinal symptoms.

## Data Availability

The data presented in this study are available on request from the corresponding author.

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
