# Peer review of "Efficacy and Safety of the Cudrania tricuspidata Extract on Functional Dyspepsia: A Randomized Double-Blind Placebo-Controlled Multicenter Study"

_jcm, 2021, doi:10.3390/jcm10225323_

Round 1

Reviewer 1 Report

This is well articulated work on Cudrania tricuspidate role in FD. The plant is a highly investigated specie with reports of important bioactivities like anti-sclerotic, anti-inflammatory, antioxidant, neuroprotective, hepatoprotective, and cytotoxic activities. The compounds responsible for the aforementioned activities have been identified. As far as the present study is concerned the results obtained are quite interesting. The manuscript is relevant to the field and well presented in a structured manner. Since the previous studies on the C. tricuspidate were mainly having in-vitro or in-vivo animal based designs, therefore, such a study was very much required. The choice of methods is appropriate and are chosen on rational scientific basis. The findings are well presented with proper figures and tables.

General concept comments

The manuscript is scientifically sound but my major concern is at the design of the study that why there has not been used a “Control” therapeutic agent that is commonly included in the design of such kind of studies? The reason for the omission may please be discussed in detail with appropriate references.

Other minor corrections/ suggestions are as follows:

  1. Line No. 59 Write the scientific name in Italic.
  2. Line No. 83 Can be given in grams. As chocolate pieces do not make any sense.
  3. Line No. 127 Kindly mention the excipients used in the formulation of the tablets. Also mention if the same excipients were present in the placebo.
  4. Line No. 274 Instead of using "adverse events" I will suggest use of "side effects" based on the outcome results of this study.
  5. Reference Nos. 15, 16, 17, 23 are incomplete. Needs a MUST correction.
  6. There are a total of 26 references out of which 14 are not current. It needs to be improved. 

Author Response

JCM-1413954

Efficacy and safety of the Cudrania tricuspidata extract on functional dyspepsia: A randomized double-blind placebo-controlled multicenter study

Thank you for giving us an opportunity to revise our manuscript with further consideration. We did our best in revising the manuscript following the comments raised by reviewers. In the revised manuscript, we highlighted revised sentences to depict what and how we made changes in the manuscript. We are pleased to submit the revised manuscript and look forward to hearing further from you. We appreciate your valuable comments.

[RESPONSE to REVIEWER 1’s COMMENT]

This is well articulated work on Cudrania tricuspidate role in FD. The plant is a highly investigated specie with reports of important bioactivities like anti-sclerotic, anti-inflammatory, antioxidant, neuroprotective, hepatoprotective, and cytotoxic activities. The compounds responsible for the aforementioned activities have been identified. As far as the present study is concerned the results obtained are quite interesting. The manuscript is relevant to the field and well presented in a structured manner. Since the previous studies on the C. tricuspidate were mainly having in-vitro or in-vivo animal based designs, therefore, such a study was very much required. The choice of methods is appropriate and are chosen on rational scientific basis. The findings are well presented with proper figures and tables.

General concept comments

The manuscript is scientifically sound but my major concern is at the design of the study that why there has not been used a “Control” therapeutic agent that is commonly included in the design of such kind of studies? The reason for the omission may please be discussed in detail with appropriate references.

[Response] As we mentioned in the introduction, functional dyspepsia has various diagnostic criteria and symptoms, it was virtually impossible to compare the efficacy with a single therapeutic agent. Therefore, the hypothesis and implementation plan of this study were conducted based on the placebo-controlled study. Since this study is the first study of CTE in human to identify the efficacy, placebo, not a therapeutic agent, was used as a control group. In further studies, the non-inferiority effect can be confirmed using the appropriate therapy agent as a control agent.

We added the limitation of this point (line 322-325) and suggest further study in conclusion (line 333-335).

Other minor corrections/ suggestions are as follows:

  1. Line No. 59 Write the scientific name in Italic.

[Response] ‘C. tricuspidata’ was rewritten in Italic.

  1. Line No. 83 Can be given in grams. As chocolate pieces do not make any sense.

[Response] We replaced the pieces of chocolate to the gram per week. It was calculated as 70 g per piece.

  1. Line No. 127 Kindly mention the excipients used in the formulation of the tablets. Also mention if the same excipients were present in the placebo.

[Response] We added the ingredients of the experimental drug and placebo in 2.3. Interventions of Method (line 132-135).

  1. Line No. 274 Instead of using "adverse events" I will suggest use of "side effects" based on the outcome results of this study.

[Response] We changed "adverse events" to "side effects" in line 271-277 and the title of table 3.

  1. Reference Nos. 15, 16, 17, 23 are incomplete. Needs a MUST correction.

[Response] We revised the references.

  1. There are a total of 26 references out of which 14 are not current. It needs to be improved. 

[Response] We rechecked all the references according to journal information.

Reviewer 2 Report

In the study, the authors tried to evaluate the effect of Cudrania tricuspidata extract (CTE) on gastrointestinal symptoms in patients with Functional dyspepsia. The study is well designed and the results presented are remarkable. The authors also highlighted the limitation of the study which is my concern, though the study is good but the number of participants/sample size is less, so will be nice if could increase the sample size, and also measure the different dose level responses. The authors should mention the complete formulation of tablet in Method section, abstract and the overall English language needs a careful check and revision.

Author Response

JCM-1413954

Efficacy and safety of the Cudrania tricuspidata extract on functional dyspepsia: A randomized double-blind placebo-controlled multicenter study

Thank you for giving us an opportunity to revise our manuscript with further consideration. We did our best in revising the manuscript following the comments raised by reviewers. In the revised manuscript, we highlighted revised sentences to depict what and how we made changes in the manuscript. We are pleased to submit the revised manuscript and look forward to hearing further from you. We appreciate your valuable comments.

[RESPONSE to REVIEWER 2’s COMMENT]

In the study, the authors tried to evaluate the effect of Cudrania tricuspidata extract (CTE) on gastrointestinal symptoms in patients with Functional dyspepsia. The study is well designed and the results presented are remarkable. The authors also highlighted the limitation of the study which is my concern, though the study is good but the number of participants/sample size is less, so will be nice if could increase the sample size, and also measure the different dose level responses.

[Response] We revised the limitation and conclusions to emphasize the need for further research to confirm the effect of CTE at the large size and various dose levels.

The authors should mention the complete formulation of the tablet in the Method section, abstract, and the overall English language needs a careful check and revision.

[Response] We added the complete formulation of the tablet in 2.3. Interventions of Method and abstract.

The English language was reconfirmed throughout the whole manuscript.

Reviewer 3 Report

The main problem with the study is that the conclusion reported that no serious adverse events were observed, and that there is a  pharmacological safety of CTE, which can be considered as a viable candidate for the management of FD. But , there is a big but... this study is very short with few samples  and ther is a lack of doise-responde data.

 Further studies are needed to assess CTE in the long-term and 
large scale dose-dependent or compared with current treatment methods.

Author Response

JCM-1413954

Efficacy and safety of the Cudrania tricuspidata extract on functional dyspepsia: A randomized double-blind placebo-controlled multicenter study

Thank you for giving us an opportunity to revise our manuscript with further consideration. We did our best in revising the manuscript following the comments raised by reviewers. In the revised manuscript, we highlighted revised sentences to depict what and how we made changes in the manuscript. We are pleased to submit the revised manuscript and look forward to hearing further from you. We appreciate your valuable comments.

[RESPONSE to REVIEWER 3’s COMMENT]

The main problem with the study is that the conclusion reported that no serious adverse events were observed, and that there is a pharmacological safety of CTE, which can be considered as a viable candidate for the management of FD. But, there is a big but... this study is very short with few samples and there is a lack of dose-response data.

 Further studies are needed to assess CTE in the long-term and large scale dose-dependent or compared with current treatment methods.

[Response] We revised the conclusion adding further study in lime 333-335.

Round 2

Reviewer 1 Report

Line No. 189 - 190 of methods section needs to be revised in accordance with the results section where "side effects" has been mentioned.

Author Response

JCM-1413954. R2.

Efficacy and safety of the Cudrania tricuspidata extract on functional dyspepsia: A randomized double-blind placebo-controlled multicenter study

Thank you for giving us an opportunity to revise our manuscript with further consideration. In this revised manuscript, we highlighted revised sentences (greenish bar) and how we made changes in the manuscript. We look forward to hearing further from you. We appreciate your valuable comments.

[RESPONSE to REVIEWER 1’s COMMENT]

Line No. 189 - 190 of methods section needs to be revised in accordance with the results section where "side effects" has been mentioned.

[Response] We changed "adverse events" to "side effects" in line 189-190. In the manuscript, we changed to “side effects”.

Reviewer 3 Report

As I explained previously the mainn problem with the  manuscript is the limited number of participants and limited sample size. In addition, there is a lack of a correct measure at  the different dose level responses.

Additional experiment shoud be done before to publish these consclusions

Author Response

JCM-1413954. R2

Efficacy and safety of the Cudrania tricuspidata extract on functional dyspepsia: A randomized double-blind placebo-controlled multicenter study

Thank you for giving us an opportunity again. In this revised manuscript, we highlighted revised sentences (greenish bar) and how we made changes in the manuscript. We look forward to hearing from you further. We appreciate your valuable comments.

[RESPONSE to REVIEWER 3’s COMMENT]

As I explained previously the main problem with the manuscript is the limited number of participants and limited sample size. In addition, there is a lack of a correct measure at the different dose level responses.

Additional experiment should be done before to publish these conclusions.

 [Response] In the power analysis to estimate the required sample size as a superiority test, at least 37 people in the experimental group were calculated as the appropriate number of participants in the study considering 1:1 allocation, a power of 80%, and a type 1 error rate (alpha) of 5% based on the result of Rodríguez, I. et al. study (Revista CENIC Ciencias Biológicas 2009, 40, 1-9.). In addition to considering a 25% drop-out rate, the total number of participants was estimated to be 100. Although it is not a large number of participants, it has been confirmed that the research design satisfies proving the efficacy and safety of CTE. Therefore, we described the evidence more clearly in the methods (line 79-82).

CTE has been found that the gastroprotective effect of the rats has a dose-dependent manner at concentrations of 5 and 10mg/kg in Kim’s study (Preventive nutrition and food science 2020, 25, 158-165). The concentrations of 10mg/kg of CTE showed a more significant reduction of gastric mucosal damage and cAMP activity to a similar extent as the Histamin-2 receptor blocker, ranitidine, than 5mg/kg of CTE. This study was conducted by converting 10mg/kg of CTE into a human equivalent dose.

This study was first conducted on humans. It was experimented with to convert the most effective concentration in animals into the human equivalent dose. We admit that there are restrictions on the inability to vary the amounts unless there is a basis for animal testing at higher concentrations. Based on additional experimental evidence, it will be implemented to confirm the effectiveness at various doses in the future.